# Intestinal permeability and peripheral immune cell composition are altered by pregnancy and adiposity at mid- and late-gestation in the mouse

**Tatiane A. Ribeiro**[1,2,3,4‡], **Jessica A. Breznik**[3,4,5,6‡], **Katherine M. Kennedy**[1,2,3,4,5,6,7,8], **Erica Yeo**[1,2], **Brianna K. E. Kennelly**[1,2], **Patrycja A. Jazwiec**[1], **Violet S. Patterson**[1,2], **Christian J. Bellissimo**[1,5], **Fernando F. Anhê**[1,2], **Jonathan D. Schertzer**[1,2], **Dawn M. E. Bowdish**[3,4,5,6,7], **Deborah M. Sloboda**[1,2,4,8,9]*

1 Department of Biochemistry and Biomedical Sciences, McMaster University, Hamilton, Ontario, Canada, 2 Farncombe Family Digestive Health Research Institute, McMaster University, Hamilton, Ontario, Canada, 3 Michael G. DeGroote Institute for Infectious Disease Research, McMaster University, Hamilton, Ontario, Canada, 4 McMaster Institute for Research on Aging, McMaster University, Hamilton, Ontario, Canada, 5 McMaster Immunology Research Centre, McMaster University, Hamilton, Ontario, Canada, 6 Department of Medicine, McMaster University, Hamilton, Ontario, Canada, 7 Firestone Institute for Respiratory Health, St. Joseph's Healthcare, Hamilton, Ontario, Canada, 8 Department of Obstetrics and Gynecology, McMaster University, Hamilton, Ontario, Canada, 9 Department of Pediatrics, McMaster University, Hamilton, Ontario, Canada

‡ TAR and JAB are joint first authors on this work.
* sloboda@mcmaster.ca

**Data Availability Statement:** We have chosen to upload our data to FigShare. Data are accessible at 10.6084/m9.figshare.22285792.

## Abstract

It is clear that the gastrointestinal tract influences metabolism and immune function. Most studies to date have used male test subjects, with a focus on effects of obesity and dietary challenges. Despite significant physiological maternal adaptations that occur across gestation, relatively few studies have examined pregnancy-related gut function. Moreover, it remains unknown how pregnancy and diet can interact to alter intestinal barrier function. In this study, we investigated the impacts of pregnancy and adiposity on maternal intestinal epithelium morphology, *in vivo* intestinal permeability, and peripheral blood immunophenotype, using control (CTL) and high-fat (HF) fed non-pregnant female mice and pregnant mice at mid- (embryonic day (E)14.5) and late (E18.5) gestation. We found that small intestine length increased between non-pregnant mice and dams at late-gestation, but ileum villus length, and ileum and colon crypt depths and goblet cell numbers remained similar. Compared to CTL-fed mice, HF-fed mice had reduced small intestine length, ileum crypt depth and villus length. Goblet cell numbers were only consistently reduced in HF-fed non-pregnant mice. Pregnancy increased *in vivo* gut permeability, with a greater effect at mid-versus late-gestation. Non-pregnant HF-fed mice had greater gut permeability, and permeability was also increased in HF-fed pregnant dams at mid but not late-gestation. The impaired maternal gut barrier in HF-fed dams at mid-gestation coincided with changes in maternal blood and bone marrow immune cell composition, including an expansion of circulating inflammatory Ly6C$^{high}$ monocytes. In summary, pregnancy has temporal effects on maternal intestinal structure and barrier function, and on peripheral immunophenotype,

**Funding:** This work was funded by a team grant from the Canadian Institutes of Health Research (CIHR) led by DMS. TAR was supported by a McMaster Institute for Research on Aging Postdoctoral Fellowship and a Michael G. DeGroote Fellowship Award. JAB was supported by a Queen Elizabeth II Scholarship in Science and Technology. KMK and EY were supported by a Farncombe Digestive Health Research Institute Student Fellowship. BKEK was supported by a CIHR Canada Graduate Scholarships – Master's (CGS-M) award. CJB is supported by the CIHR Canada Graduate Scholarships – Doctoral (CGS-D). PAJ was supported by a Thomas Neilson Scholarship, Fred & Helen Knight Enrichment Award, and an Ontario Graduate Scholarship. FFA was supported by a CIHR Fellowship. JDS holds a Canada Research Chair in Metabolic Inflammation. DMEB holds a Canada Research Chair in Aging and Immunity. DMS was funded by a Canada Research Chair in Perinatal Programming. There was no additional external funding received for this study.

**Competing interests:** The authors have declared that no competing interests exist.

which are further modified by HF diet-induced maternal adiposity. Maternal adaptations in pregnancy are thus vulnerable to excess maternal adiposity, which may both affect maternal and child health.

## Introduction

Maternal physiological, metabolic, and immunological adaptations to pregnancy are temporally regulated and integrate signals from the mother, the fetus, and the environment [1–3]. During healthy pregnancy there are gradual increases in maternal fat stores, circulating free fatty acids, glucose levels, and insulin resistance, to facilitate maternal-fetal glucose transfer and support fetal growth [4]. These changes are accompanied by immunological modifications to prevent rejection of the semi-allograft fetus, while maintaining homeostatic functions and pathogen defense [5]. However, in the context of high maternal body mass index (BMI) and excess adiposity [6], pregnancy-induced metabolic shifts are often exaggerated, characterized by pathological hyperglycemia and hyperinsulinemia at term gestation [7–11]. Excess maternal adiposity also results in adipose and placental tissue inflammation [12–17], and may alter circulating and tissue immune cell composition and function during pregnancy [13, 18–20]. This dysregulation of maternal pregnancy may be accompanied by altered intestinal function.

Maternal gut function and the intestinal microbiota change across the course of healthy pregnancy [14, 15, 21]. However, the temporal impacts of pregnancy on the gut barrier have not been previously investigated. Intestinal microbial dysbiosis and accompanying reductions in barrier function have been associated with gradual dysregulation of whole-body metabolism [22]. Indeed, changes in gut barrier function are also associated with altered peripheral immunity, as bacterial products and metabolites that cross the intestinal epithelium influence bone marrow hematopoiesis and immune cell development [23, 24]. Immune cells in turn affect metabolism during homeostasis, as well as in response to acute or chronic inflammation [23, 24].

Studies of excess adiposity and obesity in the absence of pregnancy (often conducted in males) [25], have shown associated changes within the gut, including alterations to the microbiome, impairment of intestinal barrier function, and changes to local immune cell composition with increased inflammation, which precedes expansion of adipose tissue and metabolic endotoxemia [26–29]. Mice fed a high fat (HF) diet long-term have shorter intestines as well as reduced small intestine villus length and colon crypt depth [30, 31]. We have shown in non-pregnant female mice that diet-induced obesity results in tissue- and time-dependent physiological changes within the intestines, including shortening of ileum and colon lengths and elevated paracellular permeability [32]. These changes are accompanied by alterations in peripheral metabolism, increased circulating inflammatory Ly6C$^{high}$ monocytes, and the accumulation of inflammatory macrophages in adipose tissue, contributing to metabolic dysregulation [33] Thus, loss of intestinal barrier function is likely an important contributor to local and systemic effects of HF diet intake. Despite the lack of knowledge on maternal gut function during gestation, structural and functional changes within the maternal intestinal tract likely contribute to increased pregnancy-induced energy uptake and expenditure [34–38]. Whether changes in intestinal morphology and function occur in pregnancies with a high maternal BMI is not yet clear, but effects of adiposity on whole-body pregnancy adaptations, including metabolic and immune-associated changes, could originate in the maternal gut. Indeed, we and others have reported that maternal excess adiposity/ obesity alters the composition of the

intestinal microbiome and reduces transcript levels of intestinal tight junction proteins across pregnancy [14, 15, 39, 40], and we have also observed elevated serum levels of bacterial call wall components lipopolysaccharide (LPS) and muramyl dipeptide (MDP) in HF-fed dams late in gestation [14]. In this study, we tested the hypothesis that pregnancy induces changes to the physical structure and function of the maternal intestinal epithelium, which has systemic effects on immunity, and that these adaptations are vulnerable to excess maternal adiposity.

## Materials and methods

### Animal model

Wildtype C57BL/6J virgin female mice (Jackson Laboratory, cat#000664) were bred under specific pathogen free conditions at the McMaster University Central Animal Facility. All procedures were approved by the Animal Ethics Research Board at McMaster University (AUP# 20-07-27). Mice were maintained in vent/rack cages at 22˚C on a 12-hour light/dark cycle. As summarized in Fig 1, at 5–6 weeks of age mice were randomized to receive *ad libitum* either a standard chow diet (control/CTL—17% kcal fat, 54% kcal carbohydrates, 29% kcal protein, 3 kcal/g; Harlan 8640 Teklad 22/5 Rodent Diet) or a high fat diet (HF—20% kcal protein, 20% kcal carbohydrates, 60% kcal fat, 5.21 kcal/g, Research Diets Inc. D12492). We chose a standard high fat diet, of 60% kcal fat, based on our previous work using this model in both non-pregnant female mice [32, 33] and in pregnancy [14, 15, 39].

All mice were housed 5 per cage and endpoint *in vivo* experiments in non-pregnant control fed (NP-CTL) and non-pregnant high fat fed (NP-HF) cohorts were performed after 6 weeks of diet intake. A subset of the non-pregnant female mice (after 6 weeks of CTL or HF diet feeding) were time-mated with CTL-fed C57BL/6J male mice to generate pregnancy cohorts. Briefly, two female mice were housed overnight with a single male, and successful mating was identified the next morning by the presence of a vaginal plug (designated embryonic day E0.5 of gestation) after which pregnant dams were singly housed.

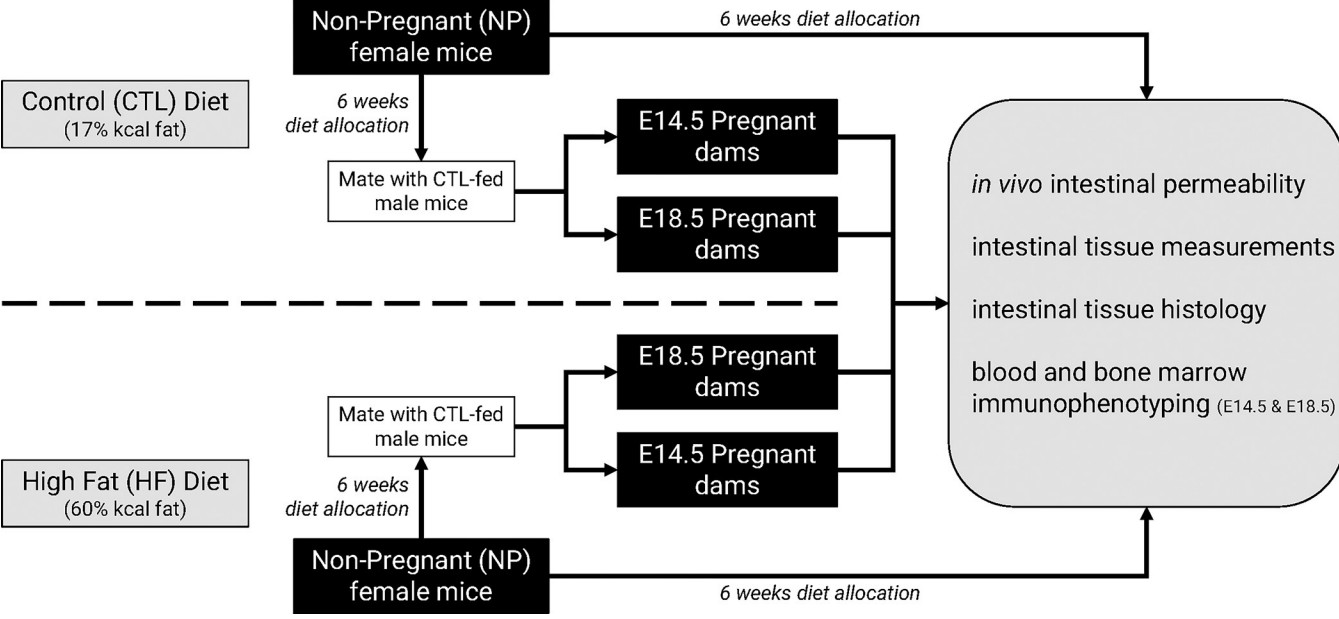

**Fig 1. Experimental design.** Schematic representation of the experimental design and animal model.

Non-pregnant female mice were randomized into two groups that received either a standard chow (CTL) or high fat (HF) diet for 6 weeks. A subset of this group underwent an *in vivo* intestinal permeability assay with subsequent intestinal tissue measurements and collection. Another subset was bred with CTL-fed males to generate pregnancy cohorts. Subsets of pregnant mice at E14.5 (mid-gestation) or E18.5 (late-gestation) were allocated for *in vivo* gut permeability assays followed by intestinal tissue measurements and collection, or assessment of peripheral immunophenotype. Mice were euthanized by cervical dislocation, or exsanguination and cervical dislocation under isoflurane anesthesia.

## *In vivo* intestinal permeability

*In vivo* gut paracellular permeability was assessed after 6 weeks of diet intake before mating (NP), and at mid- (E14.5) and late-gestation (E18.5), by measuring plasma fluorescence after oral gavage of 4-kDa FITC-dextran (Sigma-Aldrich, #46944) diluted in PBS (pH 7.4; 1.8 mM $KH_2PO_4$, 2.7 mM KCl, 10 mM $Na_2HPO_4$, 137 mM NaCl). All mice were fasted for 6 hours before the assay. Mice were gavaged with a fixed-dose of FITC-dextran (12 mg dose per mouse: 150 μL of 80 mg/mL FITC-dextran), or a weight adjusted-dose of FITC-dextran (0.5 mg/g body weight; adjusted volume of 80 mg/mL FITC-dextran). Plasma was collected at baseline (0 minutes) and 30, 60, 120, 180, and 240 minutes after gavage by centrifugation of blood collected via tail nick at 5,000 rpm for 10 minutes. Plasma was diluted 1:10 in PBS in flat-bottom 96-well black plates, and fluorescence was measured in duplicate at each time point for each mouse, using a spectrophotometer (excitation: 485 nm, emission: 530 nm; Synergy H4 Hybrid Microplate Reader, BioTek Instruments, Inc.). For each mouse, plasma fluorescence was calculated by subtracting the baseline fluorescence measurement.

## Histomorphology of intestinal tissues

To assess intestinal villus lengths, crypt depths, and goblet cell counts, tissue sections of approximately 1 cm were collected from the distal ileum and distal colon, fixed for 3 hours at room temperature in Carnoy's fixative (6:3:1 ethanol:chloroform:glacial acetic acid), and stored in 70% ethanol until processing. The tissues were processed using a Leica Biosystems processor (#TP1020) as follows: 70% ethanol (2x1 hour), 90% ethanol (2x1 hour), 100% ethanol (2x1 hour), Histoclear (Fisher Scientific, #50-329-50) (3x1 hour), and Leica Paraplast PLUS (Leica Biosystems, #39602004) (3x1 hour). Tissues were manually embedded in paraffin for cross-sectional analysis. Tissue sections of 4 μm were cut at 120 μm intervals per tissue sample. Tissue sections were deparaffinized, rehydrated and stained using Periodic Acid-Schiff (PAS) (Sigma-Aldrich, #395B-1KT). Images were captured at 10x magnification using a Nikon Eclipse NI microscope (Nikon Eclipse NI-S-E, #960122) and Nikon DS-Qi2 Colour Microscope Camera and analyzed using the ImageJ Software. Villus length and crypt depth were measured and averaged across 20 villi and 20 crypts per tissue section (3–4 tissue sections per animal). Goblet cell counts per villus and crypt were averaged across 10 villi and 10 crypts per tissue section, and across three to four tissue sections per mouse.

## Flow cytometry analysis of peripheral blood immune cells

In pregnant mice at E14.5 and E18.5, whole blood was collected in heparinized capillary tubes for analysis of circulating leukocytes including B cells, NK cells, CD4+ T cells and CD8+ T cells, monocytes (Ly6C high, Ly6C low, Ly6C-), and neutrophils, as previously described [25]. Bone marrow was extracted from femurs and prepared for surface staining as previously described [41, 42]. Stained samples were assessed with unstained and isotype controls. Samples were run on a BD Biosciences LSR II flow cytometer (BD Biosciences). Data were analyzed

using the FlowJo v9 software (Tree Star) following published gating strategies [25]. Cell counts were determined with CountBright Absolute Counting Beads (Life Technologies, #C36950).

## Statistical analysis

Each mouse is one biological replicate. Pregnancy and intestinal macrostructure measures were analyzed in R and assessed for normality by Shapiro-Wilk test (rstatix; RRID: SCR_021240) followed by t-test or Kruskal-Wallis rank sum test as appropriate (gtsummary; RRID:SCR_021319). Statistical significance was assessed either by a non-parametric Mann-Whitney, two-tailed paired t-test, or repeated measures one-way or two-way ANOVA with maternal diet and pregnancy status as factors and Tukey's post-hoc for multiple comparisons, where appropriate. Data are presented as mean ± standard error of the mean (SEM) or as box and whiskers plots min to max, where the center line indicates the median. Statistical significance was defined as $P<0.05$. Data were graphed using GraphPad Prism v9.0.

## Results

### High-fat diet consumption increases adiposity and energy consumption in females prior to and during pregnancy

Female mice were fed a standard chow diet (17% kcal fat) or a high-fat diet (60% kcal fat) for six weeks prior to and throughout pregnancy. Consistent with our previous reports using the same animal model, HF-fed females had increased body weight before and during pregnancy (S1A Fig). Energy consumption was also significantly increased in HF-fed mice before and during pregnancy (S1B Fig). Gestational weight gain was similar in CTL and HF dams (Table 1), and in accord with our previous work [14], pregnant dams had similar numbers of fetuses and reabsorptions at E14.5 and E18.5. Average fetal weight and fetal: placental weight ratio were significantly increased in HF dams compared to CTL dams at E14.5 but not E18.5 (Table 1).

### Validation of *in vivo* intestinal permeability measures in non-pregnant and pregnant mice

*In vivo* assays of intestinal barrier function often use oral gavage of fluorescently-bound non-digestible sugar molecules, such as FITC-dextran, followed by measurement of circulating fluorescence, as an indication of whole-intestine permeability [14, 43, 44]. FITC-dextran passively crosses the intestinal epithelium and thus provides a measure of paracellular

**Table 1. Maternal and fetal phenotype.**

| Characteristic | E14.5 | | | | E18.5 | | | |
|---|---|---|---|---|---|---|---|---|
| | Control[1] | HF[1] | p[1] | q[2] | Control[1] | HF[1] | p[1] | q[2] |
| **n** | **23** | **18** | | | **13** | **24** | | |
| Pregnancy weight gain (grams; median (IQR)) | 7.73 (7.04–8.48) | 9.40 (8.18–10.15) | 0.054 | 0.11 | 15.30 (13.64–16.06) | 14.90 (13.51–18.50) | 0.90 | 0.90 |
| Litter size (n) | 8 (7–9) | 9 (8–9) | 0.30 | 0.30 | 8 (7–9) | 9.00 (7.75–9.00) | 0.20 | 0.60 |
| Resorptions (n) | 0 (0–1) | 0 (0–2) | 0.20 | 0.30 | 1 (0–1) | 0 (0–1) | 0.30 | 0.60 |
| Mean fetal weight (grams; median (IQR)) | 0.23 (0.22–0.24) | 0.24 (0.23–0.25) | 0.01 | 0.059 | 1.09 (1.06–1.11) | 1.03 (1.00–1.11) | 0.40 | 0.60 |
| Mean placental weight (grams; median (IQR)) | 0.101 (0.094–0.112) | 0.097 (0.084–0.113) | 0.40 | 0.40 | 0.089 (0.089–0.093) | 0.087 (0.082–0.096) | 0.40 | 0.60 |
| Fetal:placental weight ratio | 2.24 (2.03–2.41) | 2.44 (2.24–2.90) | 0.028 | 0.083 | 12.25 (11.04–12.74) | 12.40 (10.84–13.33) | 0.60 | 0.70 |

[1]Kruskal-Wallis rank sum test

[2]False discovery rate correction for multiple testing

permeability [45, 46]. Most studies use a 150–200 µl bolus of 80 mg/ml FITC-dextran, and assess plasma concentration four hours post-gavage [14, 43, 44, 47, 48]. More recent work has highlighted the importance of considering differences in gut transit time by mouse strain and sex [45, 46, 49–52]. As maternal body weight changes dramatically over the course of gestation, and adiposity/ obesity has been reported to alter gut transit [53], and gut transit time is increased late in pregnancy [54], we set out to determine the optimal approach for assessing whole-gut permeability in CTL and HF-fed NP and pregnant female mice.

We compared plasma fluorescence at 4 hours after oral gavage of 80 mg/mL FITC-dextran at a fixed dose (12 mg) or an adjusted dose (0.5 mg/g of body weight) in NP and pregnant mice at E14.5 and E18.5 following established protocols [14, 43, 44, 47, 48]. To reduce interference of intestinal contents with transit and paracellular movement of FITC-dextran into circulation [29, 46], mice were fasted for 6 hours before gavage. Adjusted-dose bolus volumes ranged from 166.1 ± 6.0 µl in CTL-fed NP mice to 322.4 ± 32.1 µl in HF-fed E18.5 mice compared to the fixed-dose bolus volume of 150 µl (S2 Fig). We found that for both CTL and HF-fed mice, there were significant interactions between the effects of dose and pregnancy on plasma FITC fluorescence (S2B and S2C Fig). The fixed-dose generally resulted in lower plasma FITC fluorescence levels in pregnant mice (who had increased body weight) compared to non-pregnant mice, irrespective of maternal diet. However, a weight-adjusted dose did not correspond to higher plasma FITC fluorescence. Rather, adjusting the dose by body weight revealed differential effects of pregnancy on plasma fluorescence in HF-fed mice. We therefore used an adjusted dose of FITC-dextran (0.5 mg/g body weight) to investigate the effects of pregnancy and adiposity in all subsequent experiments.

## Pregnancy and high-fat diet interact to alter maternal intestinal permeability

To assess *in vivo* whole-intestine permeability using the weight-adjusted dose of FITC-dextran, we employed a time-course series of sampling over 4 hours (240 min) to ensure that the peak plasma levels of FITC fluorescence were captured. We calculated the total area under the curve (AUC) of FITC fluorescence as a measure of whole-intestine permeability. We observed that pregnancy alone had a significant effect on whole-intestine permeability (Fig 2A); pregnant mice had higher FITC fluorescence compared to non-pregnant females from 60 to 120 minutes post-gavage, and an increased AUC at E14.5. This pregnancy-induced effect on gut permeability was also seen in mice fed a HF diet (Fig 2B). HF-fed pregnant mice at mid- and late-gestation had higher plasma FITC fluorescence compared to non-pregnant HF-fed females from 30 to 60 minutes post-gavage. E14.5 dams in particular had higher plasma fluorescence at 120 and 240 minutes post-gavage and overall greater AUC (Fig 2B). These data suggest that pregnancy induces an increase in gut permeability in female mice, which is independent of diet, and that these effects are more pronounced at mid-gestation.

We next investigated the impact of diet on gut permeability in NP and pregnant female mice. Consistent with our previous findings [14], we found an increase in whole-intestine permeability in NP female HF-fed mice compared to CTL-fed mice (Fig 3A). We also observed an increase in intestinal permeability in HF pregnant mice compared to CTL mice at E14.5, but not at E18.5 (Fig 3B and 3C). Notably, in E18.5 pregnant HF-fed mice, the levels of plasma FITC fluorescence remained high at 240 min post gavage, whereas in CTL pregnant mice, the levels were comparable to NP mice. However, the overall FITC fluorescence measured by AUC between E18.5 CTL and HF dams were not different. Together these data show that whole-intestine permeability increases with pregnancy, and that this pregnancy-induced effect is further modified by diet.

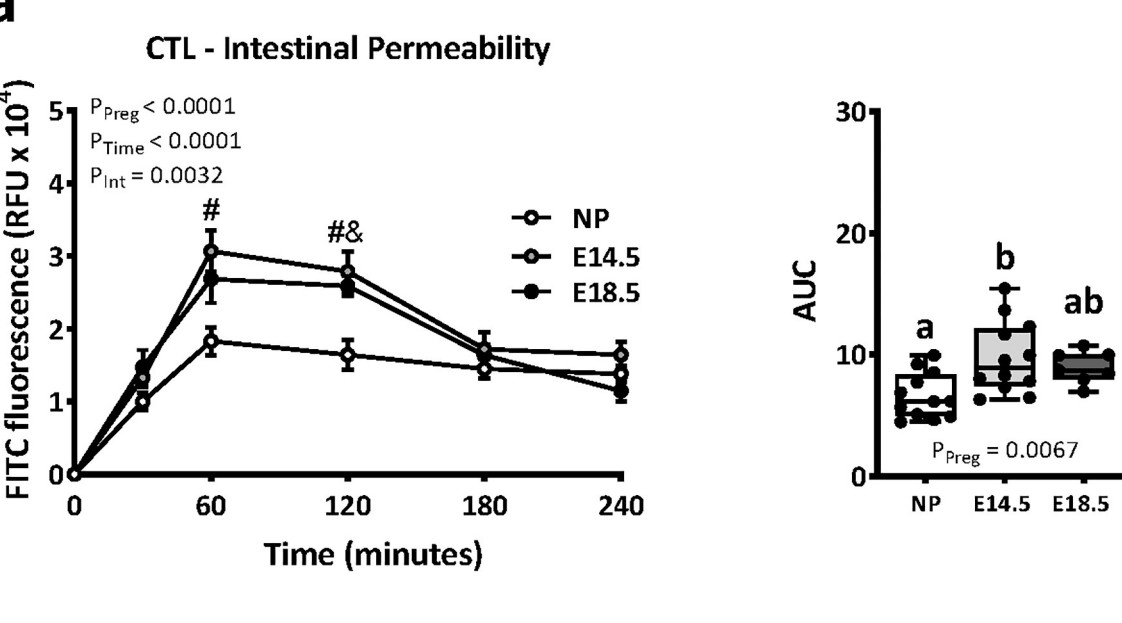

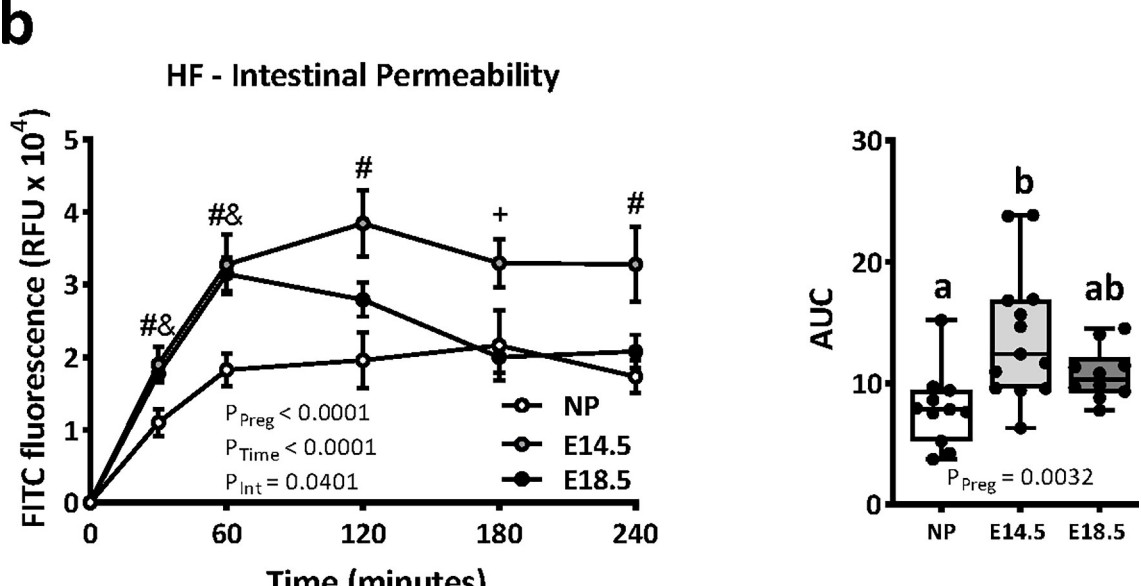

**Fig 2. Intestinal permeability increases during pregnancy and is further modified by high fat diet.** Data show plasma FITC fluorescence over 4 hours, and area under the curve (AUC), in non-pregnant (NP) and E14.5 and E18.5 pregnant female mice fed a standard diet (CTL n = 7–12) (a), or high fat diet (HF n = 9–13) (b). Data in line graphs are shown as mean ± SEM. AUC data are presented as box and whiskers plots min to max with the center line at the median and each data point is an individual mouse. For *P<0.05* multiple comparisons in line graphs: #—NP vs E14.5; &—NP vs E18.5; + E14.5 vs E18.5. Letters on AUC plots indicate statistical similarities and differences of P<0.05 between groups.

## Pregnancy and high-fat diet alter maternal intestinal structure and histomorphology

As we observed significant independent and interacting effects of pregnancy and diet on intestinal barrier function, we next investigated the ultrastructure of the intestine in non-pregnant and pregnant female mice. Total intestinal lengths increased between NP females across

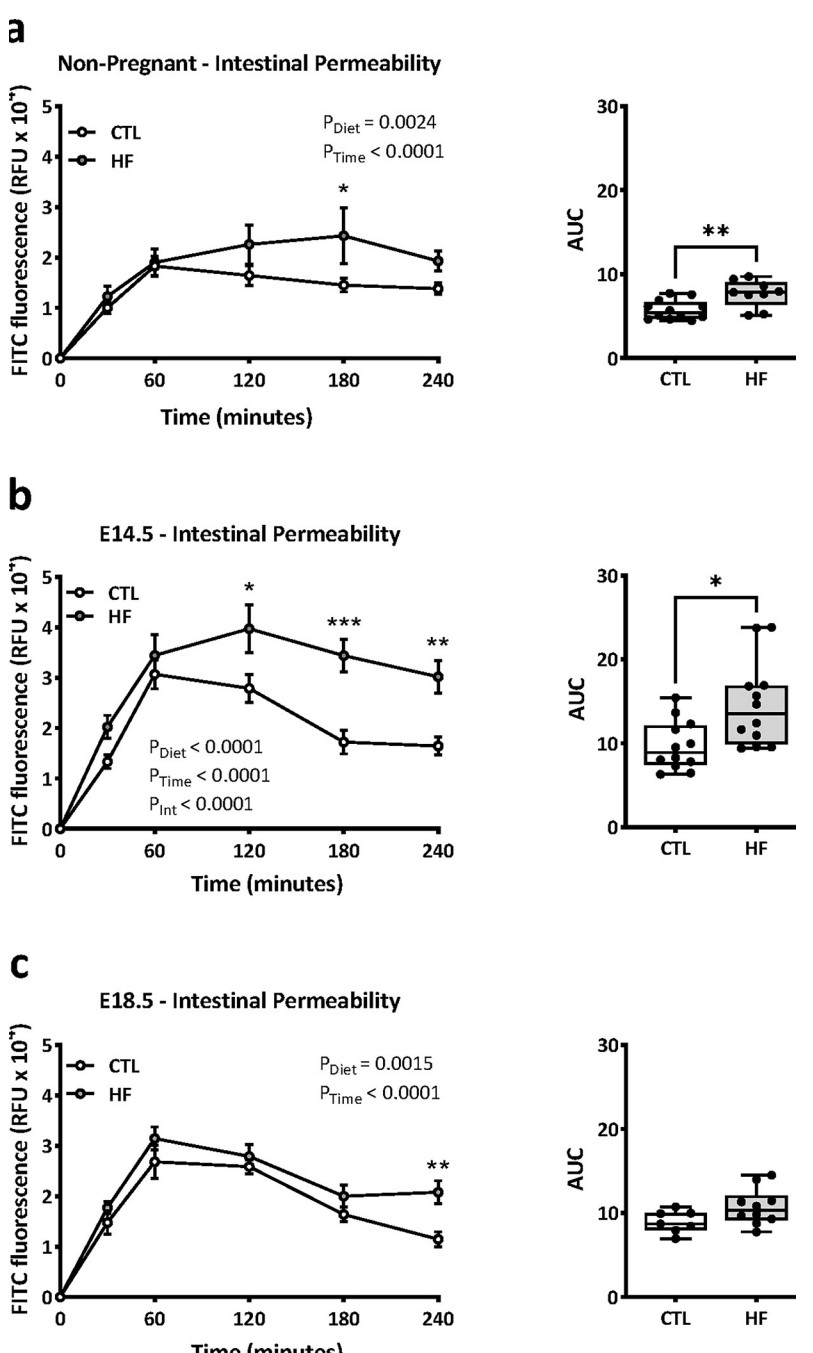

**Fig 3. Effects of a high fat diet on intestinal permeability in non-pregnant female and pregnant mice.** Data show plasma FITC-dextran over 4 hours, and area under the curve in standard chow diet (CTL n = 7–12) or high fat diet (HF n = 9–12) fed non-pregnant (NP) female mice (a), in pregnant mice at E14.5 (b), and in pregnant mice at E18.5 (c). Data in line graphs are shown as mean ± standard error of the mean (SEM). AUC data are presented as box and whiskers plots min to max where the center line indicates the median, and each data point is an individual mouse. *$P<0.05$, **$P<0.01$, ***$P<0.001$.

gestation to E18.5, and this was most pronounced in the small intestine, with an increase of over 5 cm between NP females and E18.5 dams in both CTL and HF mice. (Fig 4A–4C). There were no pregnancy-induced changes in cecal weight (Fig 4D). Consistent with other reports,

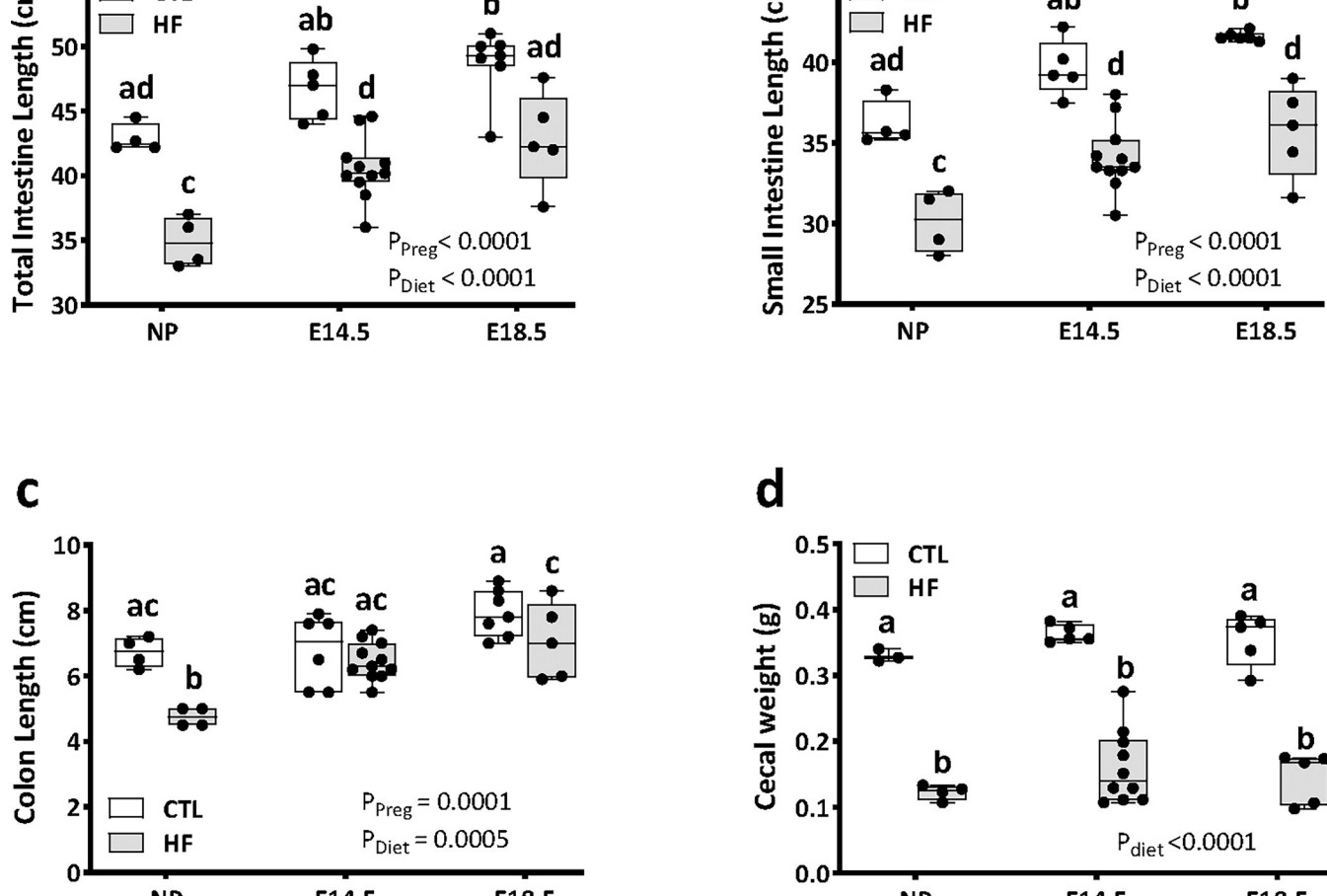

**Fig 4. High fat diet before and during pregnancy impacts intestinal length and cecal weight.** Non-pregnant female mice and pregnant mice at E14.5 and E18.5 were fed a standard chow diet (CTL n = 4–9) or high fat diet (HF n = 4–11). a) total intestine length (cm); b) small intestine length (cm); c) colon length (cm), and d) cecum weight (g) were measured at tissue collection. Data are presented as box and whisker plots min to max where the center line indicates the median, and each data point is an individual mouse. Box plots with different letters indicate statistical significance of *P<0.05*.

HF diet reduced intestinal (total, small intestine, and colon) lengths and cecal weight in NP mice (Fig 4A and 4D). HF diet similarly decreased small intestinal lengths in pregnant mice at mid- and late-gestation, but colon length was only reduced in HF dams at E18.5. Caecal weights were decreased by HF diet in pregnant dams at both E14.5 and E18.5 but were similar to those of NP HF-fed mice (Fig 4D). These data show that the gross morphology of the intestine is altered by pregnancy, particularly at E18.5, and that this effect is further modified by HF diet consumption.

Intestinal permeability and gut length have been associated with changes in villus and crypt structure [55], including changes to the mucosal barrier produced by goblet cells [56]. We next investigated the effects of pregnancy and diet on ileum and colon histomorphology. Pregnancy did not influence ileal crypt depth (Fig 5A), crypt goblet cell counts (Fig 5B), villus length (Fig 5C), or villus goblet cell numbers (Fig 5D). Consistent with other reports, HF diet significantly decreased ileal crypt depth and villus length in non-pregnant

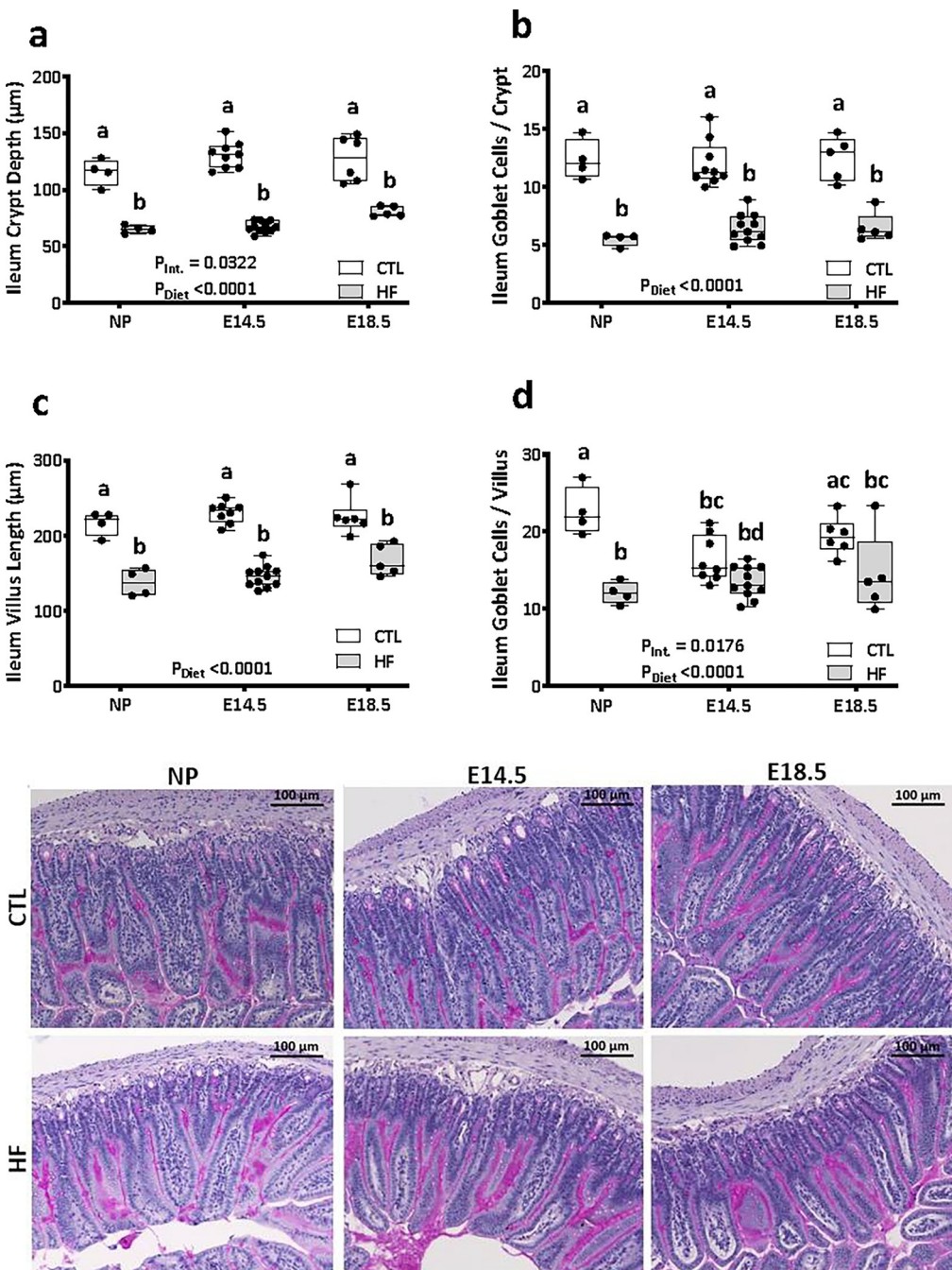

**Fig 5. High fat diet before and during pregnancy induces changes in ileal tissue structure.** Ileal tissue sections of non-pregnant and E14.5 and E18.5 pregnant mice fed a standard chow diet (CTL n = 4–9) or high fat diet (HF n = 4–11) were stained with Periodic Acid-Schiff and were used to determine: a) crypt depth; b) number of goblet cells per crypt; c) villus length; d) number of goblet cells per villus. e) representative photos of PAS staining with magnification at 10x, scale bars = 100 μm. Image analysis data in a-d are presented as box and whiskers plots min to max where the center line indicates the median, and each data point is an individual mouse. Box plots with different letters indicate statistical significance of *P<0.05*.

female mice [30] (Fig 5A and 5C), and we observed similar HF diet-induced decreases in pregnant mice at E14.5 and E18.5 (Fig 5A and 5C). HF diet also reduced goblet cell numbers in ileum villi (Fig 5D), but only in non-pregnant females. Analysis of the colon showed a main effect of diet, but not pregnancy, on colon crypt depth (Fig 6A). Colon goblet cell numbers were reduced in HF-fed non-pregnant female mice, but not during pregnancy (Fig 6B). Together, these data suggest that intestinal structure is significantly impacted by diet, with tissue-specific outcomes, and the effects of diet are further modified by pregnancy.

## High-fat diet-induced obesity alters maternal peripheral immune cell composition

We next investigated whether altered intestinal barrier function is linked to circulating immune cells, by characterizing how diet and gestational age affect peripheral blood immuno-phenotype in pregnant dams by multicolour flow cytometry. We examined the prevalence of neutrophils (Fig 7A), total monocytes (Fig 7B), B cells (Fig 7C), NK cells (Fig 7D), T cells (Fig 7E), and the ratio of CD4$^+$ to CD8$^+$ T cells (Fig 7F). Analogous to our observations of intestinal barrier function, we observed temporal effects of pregnancy on immune cell composition, which were further modified by maternal diet, and more pronounced at mid-gestation compared to late gestation. In particular, the prevalence of neutrophils was significantly decreased in HF dams at E14.5 but not E18.5, while monocyte and B cell prevalence was increased in HF dams at E14.5, but similar in CTL and HF dams at E18.5.

Given the essential role of monocyte-derived macrophages in adiposity and inflammation [57], and the significant impact we observed of HF fed on permeability at mid-gestation, we further investigated circulating monocyte populations specifically at E14.5. We observed that while total leukocyte counts remained similar in CTL and HF-fed dams (Fig 7G), total mono-cyte numbers increased (Fig 7H), especially quantities of pro-inflammatory Ly6C$^{high}$ mono-cytes (Fig 7I); for other immune cell counts see (S3 Fig). Circulating monocytes are derived from bone marrow precursors, and HF diet promotes bone marrow myelopoiesis and egress of Ly6C$^{high}$ monocytes into circulation in non-pregnant females [33]. Assessment of bone marrow in HF-fed dams compared to CTL-fed dams at E14.5 likewise showed elevated total quantities of leukocytes (Fig 7H) associated with an increased number of monocytes (Fig 7I and 7J), though the relative prevalence of Ly6C$^{high}$ monocytes was decreased in HF-fed mice compared to CTL-fed mice (Fig 7K). This suggests rapid movement of Ly6C$^{high}$ monocytes into peripheral blood. Together, these data show that maternal blood immunophenotype changes between mid- and late-gestation, and is further modified by maternal diet, with an expansion of inflammatory Ly6C$^{high}$ monocytes at mid-gestation in HF-fed dams.

## Discussion

Many individuals enter pregnancy with high body mass index (BMI) and excessive adiposity [58, 59]. It is important to understand the relationships between maternal obesity/adiposity and pregnancy adaptations, as they could affect maternal health in pregnancy and post-partum and have longer term impacts on offspring development. Indeed, changes in maternal adaptations to pregnancy may influence early life development, with consequences that extend across successive generations [60]. To our knowledge, this is the first study to assess the effects of pregnancy on *in vivo* maternal intestinal barrier function at mid-and late-gestation, as well as the modifying effects of adiposity induced by a high fat diet. We found that with advancing pregnancy there are changes in intestinal permeability and structure, as well as peripheral immune measures, and that these adaptations are further modified by maternal HF diet intake.

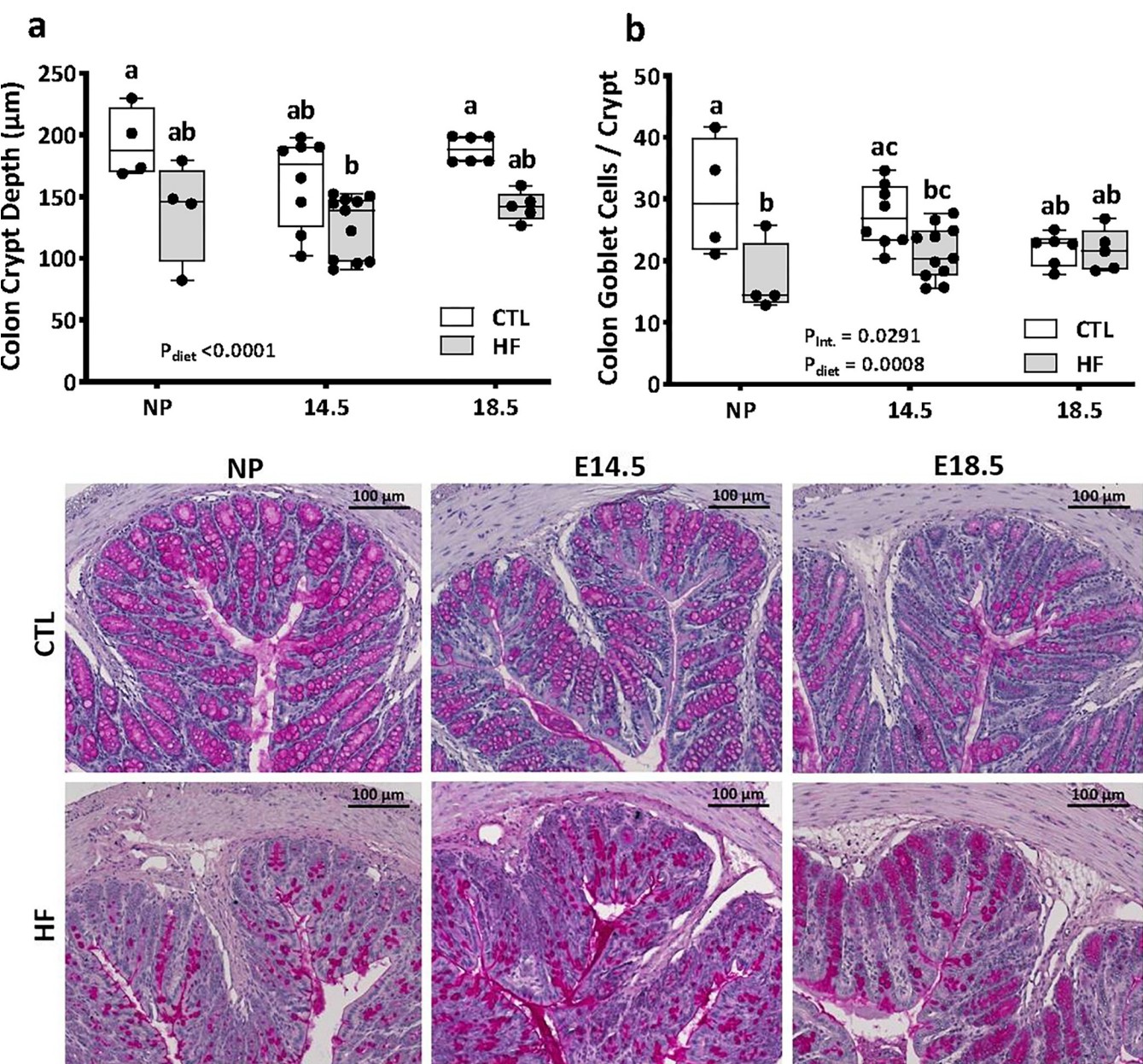

**Fig 6. High fat diet before and during pregnancy induces changes in colonic tissue structure.** Colon tissue sections of non-pregnant and E14.5 and E18.5 pregnant mice fed a standard chow diet (CTL n = 4–9) or high fat diet (HF n = 4–11) were stained with Periodic Acid-Schiff and were used to determine: a) crypt depth; b) number of goblet cells per crypt. c) representative photos of PAS staining with magnification 10x, scale bars = 100 µm. Image analysis data in a-b are presented as box and whiskers plots at the min to max where the center line indicates the median, and each data point is an individual mouse. Box plots with different letters indicate statistical significance of *P<0.05*.

Modifications to intestinal ultrastructure during pregnancy likely assist increased maternal energy demands [34–38], and this is supported by our observations of increased small intestine lengths (and thus surface area for nutrient uptake) in late-gestation in CTL dams. While one may expect an increase in villus height throughout the small intestine to augment dietary energy absorption, it has previously been reported that villus height was increased in the jejunum but not the duodenum or ileum of mice [38] and rats [61]. Our observations were

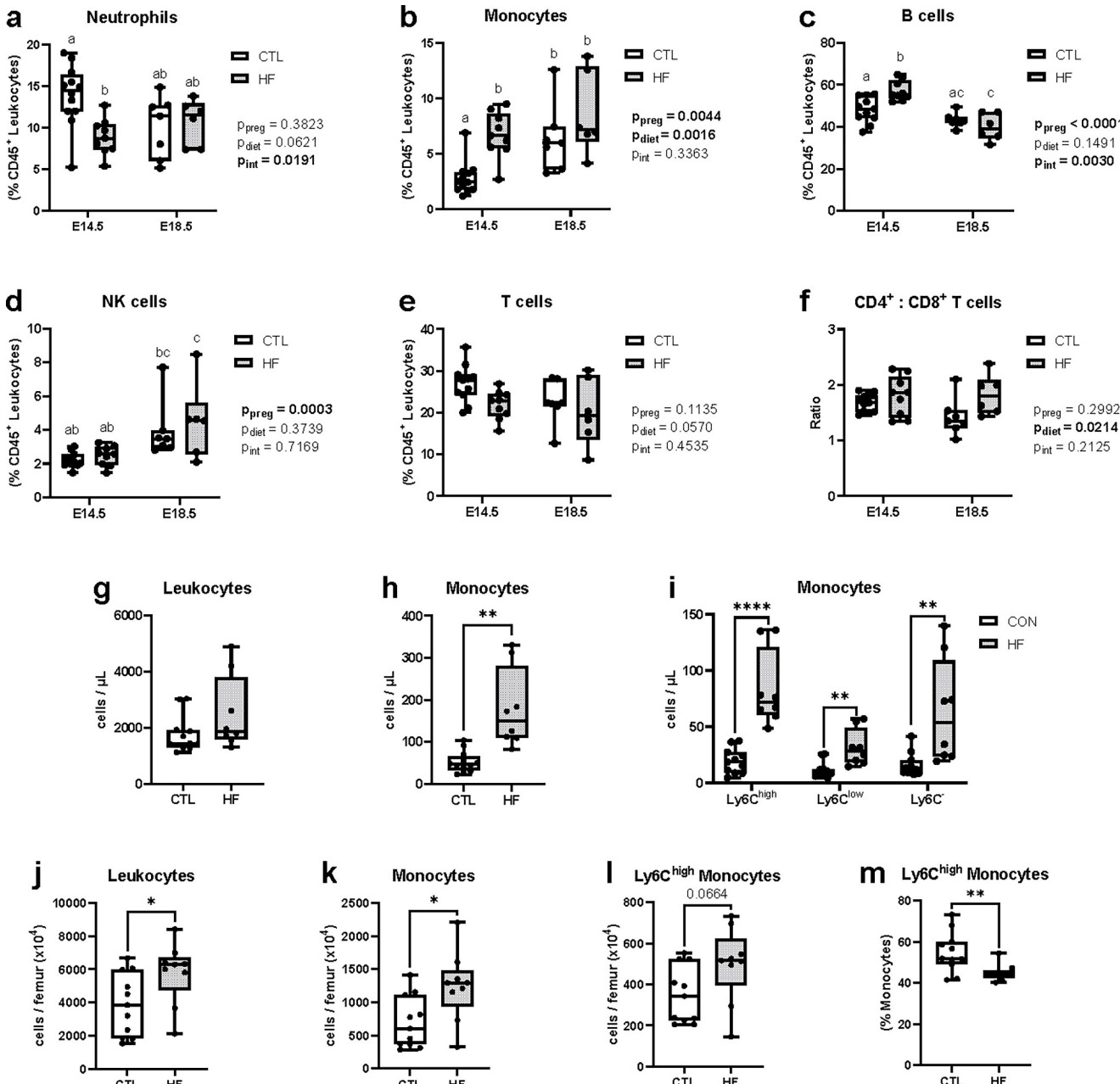

**Fig 7. Effects of maternal diet on peripheral blood and bone marrow immune cells at mid- and late-gestation.** Maternal peripheral blood and femur bone marrow immune cell populations were assessed by flow cytometry at E14.5 and/or E18.5 in dams fed a standard chow (CTL n = 7–12) or high fat (HF n = 6–12) diet. Blood prevalence (as a proportion of $CD45^+$ leukocytes) of: (a) neutrophils, (b) monocytes, (c) B cells, (d) NK cells, (e) T cells. (f) ratio of $CD4^+$ and $CD8^+$ T cells. Absolute cell counts of blood: (g) $CD45^+$ leukocytes; (h) $Ly6C^{high}$, $Ly6C^{low}$, and $Ly6C^-$ monocytes. Bone marrow absolute cell counts of: (i) leukocytes, (j) monocytes, (k) $Ly6C^{high}$ monocytes. (l) bone marrow prevalence of $Ly6C^{high}$ monocytes. Data are presented as box and whisker plots, min to max, where the centre line shows the median. Each data point indicates an individual mouse. Box plots with different letters in a-f indicate statistically significant differences of $P<0.05$. For g-l: $*P<0.05$, $**P<0.01$, $****P<0.0001$.

consistent with this finding. Likewise, pregnancy had no effect on crypt depths in either the ileum or colon. However, we observed pregnancy and diet interaction effects on ileum goblet cell numbers per villus but not crypt, and colon goblet cell number per crypt, as well as a

transient increase in total paracellular permeability at E14.5, supporting the concept that pregnancy is a dynamic process with unique spatiotemporal intestinal adaptations.

We have reported previously that HF-fed E14.5 mice had decreased transcript levels of gut tight junction proteins claudin-1 and zonulin, epithelium-secreted antimicrobial peptide β-defensin, and Muc2, the major mucin produced by goblet cells [15]. Complementing those data, here we found at mid-gestation that HF diet decreased intestinal goblet cell numbers and ileum crypt depth and villus lengths. In particular, goblet cell numbers in ileal crypts were significantly lower in HF-fed dams, and this was accompanied by an increase in whole-intestine paracellular permeability. At E18.5 we also previously reported an increase in occludin expression in HF-fed dams, but no significant changes in gut permeability [14], consistent with our observations in this study. Therefore, not all HF diet-induced changes in the gut were consistently observed across pregnancy. Our data show that mice at mid- and late gestation make different physiological adjustments in intestinal function in response to changes in the maternal environment, like diet-associated maternal adiposity, which may be necessary to maintain adequate energy uptake throughout gestation.

Increases in intestinal permeability may dictate whole-body pregnancy adaptations. Here we found that paracellular whole-intestine permeability in both CTL and HF-fed mice increased with pregnancy, but that the greatest increase in permeability was at mid-pregnancy. We have previously observed this temporal pattern in metabolic measures like fasting blood glucose and insulin, which were elevated in HF-fed dams at E14.5 but were at similar levels in CTL and HF-fed dams at E18.5 [14, 15]. It has also been reported that HF diet-induced glucose intolerance and hyperinsulinemia at mid-gestation in mice were not present in late-gestation [62]. In obese compared to lean women, increased trajectories of weight gain, insulin resistance, and circulating triglyceride levels are more apparent within the first and second trimesters of pregnancy [63–65]. We now also report significant modifications in blood immune cell composition in HF-fed dams at E14.5, but not E18.5. We have previously shown that non-pregnant female mice with diet-induced obesity have increased numbers of circulating TNF-producing inflammatory Ly6C[high] monocytes [33]. Here we similarly show that Ly6C[high] monocytes were increased in HF-fed dams at mid- but not late gestation compared to CTL dams. Studies in mice have reported lower inflammation within adipose [20, 62] or liver tissue [20] of HF-fed dams late in gestation, compared to HF-fed non-pregnant mice, and levels of circulating pro-inflammatory cytokines have been reported to decrease in women with pregravid obesity late in gestation [66]. Together these data suggest a gradual attenuation of obesity-associated inflammation, or a late-gestation adjustment of maternal immune, metabolic, and physiological traits to meet required pregnancy adaptations. In addition, though we and others have linked maternal diet with long-term effects on offspring health [67, 68] our observations of higher fetal: placental weight ratios in HF-fed dams at mid- but not late-gestation may suggest that modifications to maternal adaptations in response to excess adiposity that could be protective in fetal development.

We recognize that there are limitations to our observations that can be addressed in future studies. Despite validating the use of a body-weight-adjusted method to assess *in vivo* intestinal paracellular permeability, this approach does not provide a localized assessment of intestinal permeability, as could be measured *ex vivo* using Ussing chambers [69]. We have previously assessed serum inflammatory mediators including cytokines at E14.5 and bacterial LPS and MDP at E18.5 using the same mouse model [14, 15], but we did not repeat those measures in context of our assessments of intestinal permeability and peripheral cellular immunity within this study. Further investigations that integrate use of transcriptomic and metabolomic techniques [1–3, 70] would in addition provide insight into the impacts of maternal adiposity on intestinal, metabolic, and immune adaptations in pregnancy and the post-partum period.

In conclusion, we show that maternal epithelial barrier function and structure change over the course of pregnancy, suggesting that gut adaptations during pregnancy are dynamic. These effects are further modified by maternal diet and are accompanied by changes to maternal peripheral immune cell composition. These data extend our knowledge of maternal intestinal adaptations during pregnancy and contribute to the growing body of evidence that diet influences the intestinal environment in pregnancy and is linked with whole-body maternal health.

## Supporting information

**S1 Fig. Effects of high fat diet on maternal body weight, and energy consumption.** Female mice were fed a standard chow control (CTL; n = 7–12) diet or high fat (HF; n = 9–13) diet for 6 weeks, mated with CTL-fed mice, and maintained on their diet throughout pregnancy. (a) maternal body weight was measured at the start of diet allocation (W0), weekly (W1-W6) and during pregnancy (at gestational days E0.5, E6.5, E10.5, E14.5, E18.5); (b) maternal energy consumption (kilocalories per day). Data in line graphs are shown as mean ± SEM. *$P<0.05$.
(TIF)

**S2 Fig. Validation of FITC-dextran dose.** Validation experiments were performed to assess the appropriate dose of FITC-dextran by oral gavage to evaluate *in vivo* intestinal permeability in non-pregnant female mice, and pregnant E14.5 and E18.5 dams, fed a standard chow diet (CTL; n = 7–12) or high fat diet (HF; n = 9–14). a) volume of FITC -dextran administered in fixed-dose compared to weight adjusted-dose; b) plasma FITC fluorescence in CTL diet-fed non-pregnant female mice and pregnant dams at 4 hours post gavage; c) plasma FITC fluorescence in HF-fed non-pregnant female mice and pregnant dams at 4 hours post gavage. Data are shown as scatter dot plots and the center line indicates the median. Each data point is an individual mouse. Box plots with different letters indicate statistical significance of $P<0.05$.
(TIF)

**S3 Fig. Effects of HF diet on peripheral blood immune cell numbers at E14.5.** Maternal peripheral blood and immune cell populations of standard chow control-fed (CTL; n = 11–12) dams and high fat-fed (HF; n = 9) dams were assessed by flow cytometry at E14.5. Absolute cell counts of: (a) neutrophils, (b) B cells, (c) NK cells, and (d) T cells. Each data point indicates an individual mouse. Data are presented as box and whisker plots, min to max, where the centre line shows the median. *$P<0.05$.
(TIFF)

## Acknowledgments

We would like to thank the McMaster University Central Animal Facility for their care of the animals and Hong Liang and the McMaster Immunology Research Centre flow cytometry facility.

## Author Contributions

**Conceptualization:** Tatiane A. Ribeiro, Jessica A. Breznik, Deborah M. Sloboda.

**Formal analysis:** Tatiane A. Ribeiro, Jessica A. Breznik, Katherine M. Kennedy.

**Funding acquisition:** Dawn M. E. Bowdish, Deborah M. Sloboda.

**Investigation:** Tatiane A. Ribeiro, Jessica A. Breznik, Deborah M. Sloboda.

**Methodology:** Tatiane A. Ribeiro, Jessica A. Breznik, Deborah M. Sloboda.

**Project administration:** Tatiane A. Ribeiro, Deborah M. Sloboda.

**Resources:** Dawn M. E. Bowdish, Deborah M. Sloboda.

**Supervision:** Dawn M. E. Bowdish, Deborah M. Sloboda.

**Validation:** Tatiane A. Ribeiro.

**Visualization:** Tatiane A. Ribeiro, Jessica A. Breznik.

**Writing – original draft:** Tatiane A. Ribeiro, Jessica A. Breznik, Deborah M. Sloboda.

**Writing – review & editing:** Tatiane A. Ribeiro, Jessica A. Breznik, Katherine M. Kennedy, Erica Yeo, Brianna K. E. Kennelly, Patrycja A. Jazwiec, Violet S. Patterson, Christian J. Bellissimo, Fernando F. Anhê, Jonathan D. Schertzer, Dawn M. E. Bowdish, Deborah M. Sloboda.

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
