## [Decision Letter · Decision Letter 0]

8 Mar 2023

PONE-D-22-25312Intestinal permeability and peripheral immune cell composition are altered by pregnancy and adiposity at mid- and late-gestation in the mousePLOS ONE

Dear Dr. Sloboda,

Thank you for submitting your manuscript to PLOS ONE. After careful consideration, we feel that it has merit but does not fully meet PLOS ONE’s publication criteria as it currently stands. Therefore, we invite you to submit a revised version of the manuscript that addresses the points raised during the review process.

We look forward to receiving your revised manuscript.

Kind regards,

Christopher Torrens

Academic Editor

PLOS ONE

Journal Requirements:

“This work was funded by a team grant from the Canadian Institutes of Health Research (CIHR) led by DMS. TAR was supported by a McMaster Institute for Research on Aging Postdoctoral Fellowship and a Michael G. DeGroote Fellowship Award. JAB was supported by a Queen Elizabeth II Scholarship in Science and Technology. Work in the Bowdish laboratory is supported by the McMaster Immunology Research Centre and the M. G. DeGroote Institute for Infectious Disease Research. DMEB and DMS are/were supported by the CIHR and are Canada Research Chairs.”

Reviewers' comments:

Reviewer's Responses to Questions

**Comments to the Author**

1. Is the manuscript technically sound, and do the data support the conclusions?

Reviewer #1: Yes

Reviewer #2: Yes

Reviewer #3: Yes

2. Has the statistical analysis been performed appropriately and rigorously? 

Reviewer #1: Yes

Reviewer #2: Yes

Reviewer #3: Yes

3. Have the authors made all data underlying the findings in their manuscript fully available?

Reviewer #1: Yes

Reviewer #2: Yes

Reviewer #3: Yes

4. Is the manuscript presented in an intelligible fashion and written in standard English?

Reviewer #1: Yes

Reviewer #2: Yes

Reviewer #3: Yes

5. Review Comments to the Author

Reviewer #1: This is an interesting study by Ribeiro and colleagues, who have examined the structural morphology of mouse intestine under normal diet and high fat conditions at different stages in pregnancy This is a well written and well designed study with interesting observations on the dynamic changes in intestinal structure and permeability under the experimental conditions.

I have raised a couple of points below, which should be addressed.

Abstract.

Provide statement on the inferences from the findings – why are they important (currently, there is only a summary of the data).

Introduction

Place rework the sentence ‘This barrier is necessary to prevent systemic dissemination of luminal contents including the intestinal microbiota.’ It suggests total breakdown of the gut barrier and opposed to more subtle modulation of permeability.

The sentence ‘Thus, maintenance of the intestinal barrier is essential to prevent both local

and systemic effects of HF diet intake.’ suggests a ‘cause and effect’ relationship between gut permeability and body-wise effects of a diet high in fat. While it may be part of the story, there are other factors involved too, which should be acknowledged.

Reviewer #2: This study verified that pregnancy induced changes in the physical structure and function of maternal intestinal epithelium, which had a systemic effect on immunity, and maternal obesity caused by the HF diet further aggravated this effect. Therefore, I think this study is of great significance and suggest that it be revised and published.

1、It is not necessary to introduce statistical methods below each figure, except for special methods

2、Inflammatory factors in serum have a close relationship with intestinal permeability. It is suggested that some inflammatory factors be supplemented in serum.

3、Some figures are marked as ' * ', some are marked as ' abc ', and some are not marked.It is suggested to use a unified symbol to mark the significance, and add that there is no significant difference if there is no mark.

4、It is recommended that ' ab ' represents P < 0.05 and ' AB ' represents P < 0.01.

5、The abstract is a high-level summary of the full text.Although there is no clear definition, the general abstract must contain key words, and 'maternal adaptation' does not appear in the abstract.

Reviewer #3: This well-written manuscript describes novel data that addresses a gap in the current literature at the intersection of gut structure and function, high fat diets, and pregnancy. T

Questions and concerns that might strengthen the manuscript.

• In the introduction (line 77), the author references unpublished data which does not appear to be appropriately cited.

• In the methods (line 102), there is no justification for the use of a 60% HF diet as opposed to using a 45% HF diet,which in the literature appears to be more physiologic for translation to humans. Exclusion of an intermediate HF diet, 45% HF diet, also limits the opportunity for exploring dose-dependent responses.

• In the methods (line 114), there is no justification for why the gestational timepoints of E14.5 and E18.5 were chosen.

• In the methods (line 120), the use of the FITC assay appears to cloud the picture given high doses of the biomarkers were needed to illicit a response in the pregnant cohort. This in part may be due to known dysmotility caused by relaxin during pregnancy leading to slower transit in the gut. High fat diets have also been shown to slow down motility in the gut and may play a role in the varied results of this assay. As a future approach, the authors might consider ex vivo Ussing chamber permeability studies, which will not be clouded by differing body weights or altered motility.

• It would be interesting to know if the pups were weighed from each group and if differences were seen.

• In the discussion, the inclusion of study limitations and relevance to human studies is not discussed.

6. PLOS authors have the option to publish the peer review history of their article (what does this mean?). If published, this will include your full peer review and any attached files.

Reviewer #1: No

Reviewer #2: No

Reviewer #3: No

---

## [Author Response · Author response to Decision Letter 0]

27 Mar 2023

Reviewer #1: This is an interesting study by Ribeiro and colleagues, who have examined the structural morphology of mouse intestine under normal diet and high fat conditions at different stages in pregnancy This is a well written and well designed study with interesting observations on the dynamic changes in intestinal structure and permeability under the experimental conditions. Abstract. Provide statement on the inferences from the findings – why are they important (currently, there is only a summary of the data). We thank the reviewer for the suggestion and have inserted text within the abstract stating: “Moreover, it remains unknown how pregnancy and diet can interact to alter intestinal barrier function.” (lines 32-33) and “Maternal adaptations in pregnancy are thus vulnerable to excess maternal adiposity, which may affect maternal and child health.” (lines 48-49)

Introduction Place rework the sentence ‘This barrier is necessary to prevent systemic dissemination of luminal contents including the intestinal microbiota.’ It suggests total breakdown of the gut barrier and opposed to more subtle modulation of permeability. This sentence has been rewritten (see lines 67-69): “Intestinal microbial dysbiosis and accompanying reductions in barrier function have been associated with gradual dysregulation of whole-body metabolism.” A new reference – Lam et al. 2012 has been added (https://doi.org/10.1371/journal.pone.0034233

The sentence ‘Thus, maintenance of the intestinal barrier is essential to prevent both local and systemic effects of HF diet intake.’ suggests a ‘cause and effect’ relationship between gut permeability and body-wise effects of a diet high in fat. While it may be part of the story, there are other factors involved too, which should be acknowledged. We thank the reviewer for the suggestion. We have revised the sentence so it is less ‘cause and effect’ focussed but still provides a summary of the preceding information on intestinal barrier function and local and systemic changes that are observed in diet-induced obesity (See lines 85-86): "Thus, loss of intestinal barrier function is likely an important contributor to local and systemic effects of HF diet intake.”

Reviewer #2: This study verified that pregnancy induced changes in the physical structure and function of maternal intestinal epithelium, which had a systemic effect on immunity, and maternal obesity caused by the HF diet further aggravated this effect. Therefore, I think this study is of great significance and suggest that it be revised and published. 1、It is not necessary to introduce statistical methods below each figure, except for special methods The figure caption text has been revised as requested to remove statistical methods as they are described fully within the methods. (See lines 25-255; 274-277; 299-301; 324-326; 335-337; 376-381; 679-681; 691-694; 701-703).

Reviewer #2: This study verified that pregnancy induced changes in the physical structure and function of maternal intestinal epithelium, which had a systemic effect on immunity, and maternal obesity caused by the HF diet further aggravated this effect. Therefore, I think this study is of great significance and suggest that it be revised and published. 1、It is not necessary to introduce statistical methods below each figure, except for special methods The figure caption text has been revised as requested to remove statistical methods as they are described fully within the methods. (See lines 25-255; 274-277; 299-301; 324-326; 335-337; 376-381; 679-681; 691-694; 701-703). 2、Inflammatory factors in serum have a close relationship with intestinal permeability. It is suggested that some inflammatory factors be supplemented in serum. We have interpreted ‘supplemented’ as ‘measured’. We agree with the reviewer that intestinal permeability has been associated with systemic inflammation, which may be reflected in the composition of serum inflammatory factors like cytokines, and/or cellular components of immunity like monocytes.

Using the same high fat diet model in pregnancy, we have previously measured (ref. 23) serum cytokines at E14.5 in chow and high-fat fed dams (see below from Figure 4 of Wallace et al., 2019: 10.1038/s41598-019-54098-x). We detected no statistically significant differences in levels of these serum inflammatory factors between chow and HF-fed dams.

However, we previously observed that levels of bacterial products LPS (lipopolysaccharide) and MDP (muramyl dipeptide) were elevated in serum of HF-fed dams compared to chow-fed dams at E18.5 (using the same diet model) (see right and Figure 4 of Gohir & Kennedy et al., 2019: https://doi.org/10.1113/JP277353).

We realize that these data were not explicitly cited in the current manuscript, so text has been added as follows in the introduction as part of the justification for this study: Lines 95-96: “..., and we have also observed elevated serum levels of bacterial call wall components lipopolysaccharide (LPS) and muramyl dipeptide (MDP) in HF-fed dams late in gestation (14).”

The reference for Gohir & Kennedy et al., 2019 (https://doi.org/10.1113/JP277353) was included. We have also acknowledged that we did not measure serum cytokines or bacterial products in this particular manuscript within a new section on study limitations within the discussion, as requested by Reviewer 3 (lines 443-447), and referenced both of the studies above. “…We have previously assessed serum inflammatory mediators including cytokines at E14.5 and bacterial LPS and MDP at E18.5 using the same mouse model (14, 23), but we did not repeat those measures in context of our assessments of intestinal permeability and peripheral cellular immunity within this study…”.

3、Some figures are marked as ' * ', some are marked as ' abc ', and some are not marked. It is suggested to use a unified symbol to mark the significance, and add that there is no significant difference if there is no mark. In the figures in this manuscript, statistical significance between groups is indicated by asterisks when two-group analyses i.e., Student’s t test or Mann-Whitney U test, were used. In contrast, two-way ANOVA analyses are reported with main effects and post-hoc similarities and differences are indicated by letters. If two groups share the same letters, then they are similar, whereas if they have different letters, they are significantly different (i.e., P is at least <0.05). We regrettably cannot assign P values based on letter combinations, as statistically significant differences may not be the same between dissimilar letter pairs. We believe the current formatting is not inconsistent with journal requirements but would be happy to change the formatting if the Editor deems it appropriate.

4、It is recommended that ' ab ' represents P < 0.05 and ' AB ' represents P < 0.01. Please see the above response for comment 3. 5、The abstract is a high-level summary of the full text. Although there is no clear definition, the general abstract must contain key words, and 'maternal adaptation' does not appear in the abstract.

We thank the reviewer for this observation. We have included “maternal adaptation” in the abstract (line 31): “Despite significant physiological maternal adaptations that occur across gestation…”

Reviewer #3: This well-written manuscript describes novel data that addresses a gap in the current literature at the intersection of gut structure and function, high fat diets, and pregnancy. T Questions and concerns that might strengthen the manuscript. • In the introduction (line 82), the author references unpublished data which does not appear to be appropriately cited. This data referenced was included within a manuscript that was in review at the time of submission of this manuscript. That manuscript has since been accepted for publication. The appropriate reference (see below) has been added (line 82 – ref. 32). Breznik JA, Jury J, Verdu EF, Sloboda DM, Bowdish DME. Diet-induced obesity alters intestinal monocyte-derived and tissue-resident macrophages and increases intestinal permeability in female mice independent of TNF. Am J Physiol Gastrointest Liver Physiol. 2023 Feb 7. doi: 10.1152/ajpgi.00231.2022. Epub ahead of print. PMID: 36749921.

• In the methods (line 102), there is no justification for the use of a 60% HF diet as opposed to using a 45% HF diet, which in the literature appears to be more physiologic for translation to humans. Exclusion of an intermediate HF diet, 45% HF diet, also limits the opportunity for exploring dose-dependent responses. We thank the reviewer for this observation, and we agree that adding context for choosing a 60% HF diet is important. We chose that diet because this study builds on our previous work that has used that diet to obtain a phenotype of excess adiposity/obesity with accompanying metabolic dysregulation. We recognize that there has been discussion in the literature that a 45% kcal from fat diet may be more physiologically translatable to humans, and agree with the reviewer that including an ‘intermediate’ high fat diet would offer opportunities for exploring dose-dependent responses. Currently we are conducting studies using both the 45% and 60% diets.

We would also like to point out that the commonly used 45% / 60% diets (from ResearchDiets) have differences in relative dietary fat composition (i.e., there is more soybean oil of total fat in the 60% diet) and sucrose content. Even studies that attempt to identify the ‘ideal’ HF diet recognize that effects of HF diet with specific mouse models may differ. Mouse strain, sex, age, and health status affect the response of mice to obesogenic diets, and the composition of the fat and carbohydrate content, and macronutrient and micronutrient components within the diet can also affect the development of obesity and other effects of high fat diet intake (see https://doi.org/10.1186/s13098 021 00647 2 and https://doi.org/10.1038/oby.2007.60 8 )). Thus, understanding specific effects of HF diet in a specific lab setting and mouse model type, while acknowledging limitations, is essential to develop ways to manipulate that model. We have added the following text into the Methods, which we hope addresses the Reviewer’s concerns (lines 110-112):

We chose a standard high fat diet, of 60% kcal fat, based on our previous work using this model in both non-pregnant female mice (32, 33) and in pregnancy (14, 15, 40).” References:

14. Gohir W, Kennedy KM, Wallace JG, Saoi M, Bellissimo CJ, Britz-McKibbin P, et al. High-fat diet intake modulates maternal intestinal adaptations to pregnancy and results in placental hypoxia, as well as altered fetal gut barrier proteins and immune markers. The Journal of physiology. 2019;597(12):3029-51.

15. Wallace JG, Bellissimo CJ, Yeo E, Fei Xia Y, Petrik JJ, Surette MG, et al. Obesity during pregnancy results in maternal intestinal inflammation, placental hypoxia, and alters fetal glucose metabolism at mid-gestation. Scientific reports. 2019;9(1):17621. 32. Breznik JA, Jury J, Verdu EF, Sloboda DM, Bowdish DME. Diet-induced obesity alters intestinal monocyte-derived and tissue-resident macrophages and increases intestinal permeability in female mice independent of TNF. Am J Physiol Gastrointest Liver Physiol. 2023 Feb 7. doi: 10.1152/ajpgi.00231.2022. Epub ahead of print. PMID: 36749921.

33. Breznik JA, Foley KP, Maddiboina D, Schertzer JD, Sloboda DM, Bowdish DME. Effects of Obesity-Associated Chronic Inflammation on Peripheral Blood Immunophenotype Are Not Mediated by TNF in Female C57BL/6J Mice. ImmunoHorizons. 2021;5(6):370-83.

40. Gohir W, Whelan FJ, Surette MG, Moore C, Schertzer JD, Sloboda DM. Pregnancy-related changes in the maternal gut microbiota are dependent upon the mother's periconceptional diet. Gut microbes. 2015;6(5):310-20.

• In the methods (line 114), there is no justification for why the gestational timepoints of E14.5 and E18.5 were chosen. We thank the reviewer for this observation. Within the abstract (lines 33-34) we identified that in C57BL/6 mouse models of pregnancy, embryonic day (E)14.5 is mid-gestation, while E18.5 is late/term-gestation. Pregnancy is typically <20 days in C57BL/6 mice. We recognize that this was not stated within the Methods, and have amended text (highlighted) within the methods (line 125): “Subsets of pregnant mice at E14.5 (mid-gestation) or E18.5 (late-gestation) were allocated for…”

• In the methods (line 120), the use of the FITC assay appears to cloud the picture given high doses of the biomarkers were needed to illicit a response in the pregnant cohort. This in part may be due to known dysmotility caused by relaxin during pregnancy leading to slower transit in the gut. High fat diets have also been shown to slow down motility in the gut and may play a role in the varied results of this assay. As a future approach, the authors might consider ex vivo Ussing chamber permeability studies, which will not be clouded by differing body weights or altered motility. We believe we have addressed the Reviewer’s concerns about FITC-dextran assays through our validation of a new approach for the assay that considers dysmotility and differences in body weight by using a time course and dose adjusted by body weight. As stated within the manuscript (lines 208-209): “…recent work has highlighted the importance of considering differences in gut transit time by mouse strain and sex (46, 47, 50-53). As maternal body weight dramatically over the course of gestation, and adiposity/ obesity has been reported to alter gut transit (54), and gut transit time is increased late in pregnancy (55), we set out to determine the optimal approach for assessing whole-gut permeability in CTL and HF-fed NP and pregnant female mice.” Our validation experiments showed that a fixed dose, as suggested by the reviewer, was inappropriate for the above reasons, and led us to use an adjusted dose of FITC-dextran by body weight. We fasted mice for 6 hours prior to the assay and our experiments incorporated sampling over 4 hours to assess if peaks in fluorescence were different over time (changes in time to peak fluorescence between animal groups may be associated with differences in gut motility). Therefore, we did consider both differing body weights and altered motility in our experimental approach for the in vivo permeability assay. The final dose of FITC-dextran that we used was 0.5 mg/g body weight (lines 215-217), which is lower than that of other publications (e.g. 0.6 mg/g body weight used by Woting & Blaut, 2018; 10.3390/nu10060685).

However, we agree with the reviewer that ex vivo Ussing chamber experiments are the ‘gold standard’ to measure tissue-localized intestinal permeability. We have previously published data on in vivo permeability using a fixed dose of 150 μL or 200 μL of 80mg/mL FITC-dextran, with corroboration of those data using Ussing chambers. Specifically, we found using a fixed FITC-dextran dose and Ussing chambers that old mice have greater intestinal permeability (specifically localized to the colon) than young mice (see figure below and Figure 3 in Thevaranjan et al., 2018 10.1016/j.chom.2018.03.006):

We also found using a fixed FITC-dextran dose and Ussing chambers that in non-pregnant female mice a high fat diet increases intestinal permeability within 6 weeks diet allocation, with effects also apparent after 18 weeks (see figure below and Figure 1 in Breznik et al., 2023 10.1152/ajpgi.00231.2022):

While Ussing chamber experiments were outside the scope of the current study, we are incorporating both in vivo and ex vivo assessments of permeability into further studies. For the Reviewer and Editor’s interest, we have included below unpublished data from a recent study examining changes to the gut in lactation and effects of material diet. We have performed in vivo permeability assessments with an adjusted dose of FITC-dextran by body weight, as in the current manuscript, with confirmation of our observations with ex vivo Ussing chamber assessments:

We believe that these data collectively indicate that our results in the current manuscript are not confounded by an increased dose of FITC-dextran. Furthermore, these data indicate that observations of increased permeability by in vivo assessments by FITC-dextran gavage align well with ex vivo paracellular permeability assessments using Ussing chambers, irrespective of whether a fixed or adjusted dose is used. However, from the observations of validation assessments that were performed in the current manuscript, we believe our adjusted dose approach is superior for comparisons of in vivo permeability when differing body weights or altered gastrointestinal motility may be present. To acknowledge the Reviewer’s suggestion, we have added text discussing limitations of our approach and recommend the use of Ussing chambers for localized assessments of intestinal permeability (lines 441-443): “Despite validating the use of a body-weight-adjusted method to assess in vivo intestinal paracellular permeability, this approach does not provide a localized assessment of intestinal permeability, as could be measured ex vivo using Ussing chambers (70).”

Reference 70: Thomson A, Smart K, Somerville MS, Lauder SN, Appanna G, Horwood J, et al. The Ussing chamber system for measuring intestinal permeability in health and disease. BMC gastroenterology. 2019;19(1):98. • It would be interesting to know if the pups were weighed from each group and if differences were seen.

We describe this result on lines 193-194:

“Average fetal weight and fetal: placental weight ratio were significantly increased in HF dams compared to CTL dams at E14.5 but not E18.5 (Table 1).”

• In the discussion, the inclusion of study limitations and relevance to human studies is not discussed. We thank the reviewer for this suggestion, and have added the following text to the discussion (lines 440-447):

“We recognize that there are limitations to our observations that can be addressed in future studies. Despite validating the use of a body-weight-adjusted method to assess in vivo intestinal paracellular permeability, this approach does not provide a localized assessment of intestinal permeability, as could be measured ex vivo using Ussing chambers (70). We have previously assessed serum inflammatory mediators including cytokines at E14.5 and bacterial LPS and MDP at E18.5 using the same mouse model (14, 15), but we did not repeat those measures in context of our assessments of intestinal permeability and peripheral cellular immunity within this study.” References:

14. Gohir W, Kennedy KM, Wallace JG, Saoi M, Bellissimo CJ, Britz-McKibbin P, et al. High-fat diet intake modulates maternal intestinal adaptations to pregnancy and results in placental hypoxia, as well as altered fetal gut barrier proteins and immune markers. The Journal of physiology. 2019;597(12):3029-51.

15. Wallace JG, Bellissimo CJ, Yeo E, Fei Xia Y, Petrik JJ, Surette MG, et al. Obesity during pregnancy results in maternal intestinal inflammation, placental hypoxia, and alters fetal glucose metabolism at mid-gestation. Scientific reports. 2019;9(1):17621.

70. Thomson A, Smart K, Somerville MS, Lauder SN, Appanna G, Horwood J, et al. The Ussing chamber system for measuring intestinal permeability in health and disease. BMC gastroenterology. 2019;19(1):98.

---

## [Editor Report · Decision Letter 1]

13 Apr 2023

Intestinal permeability and peripheral immune cell composition are altered by pregnancy and adiposity at mid- and late-gestation in the mouse

PONE-D-22-25312R1

Dear Dr. Sloboda,

We’re pleased to inform you that your manuscript has been judged scientifically suitable for publication and will be formally accepted for publication once it meets all outstanding technical requirements.

Kind regards,

Christopher Torrens

Academic Editor

PLOS ONE
---

## [Editor Report · Acceptance letter]

4 May 2023

PONE-D-22-25312R1 

Intestinal permeability and peripheral immune cell composition are altered by pregnancy and adiposity at mid- and late-gestation in the mouse 

Dear Dr. Sloboda:

I'm pleased to inform you that your manuscript has been deemed suitable for publication in PLOS ONE. Congratulations! Your manuscript is now with our production department. 

Kind regards, 

on behalf of

Dr. Christopher Torrens 

Academic Editor

PLOS ONE